# Ancestral origins of *TYR* and *OCA2* gene mutations in oculocutaneous albinism from two admixed populations in Colombia

Wilmer A. Cárdenas[1], Andrew B. Conley[2], Shashwat Deepali Nagar[2,3], Diana L. Núñez-Ríos[4]*, I. King Jordan[2,3], María Claudia Lattig[1]

1 Facultad de Ciencias, Universidad de Los Andes, Bogotá, Colombia, 2 Applied Bioinformatics Laboratory, Atlanta, Georgia, United States of America, 3 School of Biological Science, Georgia Institute of Technology, Atlanta, Georgia, United States of America, 4 Division of Human Genetics, Department of Psychiatry, Yale University School of Medicine, New Haven, CT, United States of America

* diana.nunez@yale.edu

**Data Availability Statement:** All relevant data are within the manuscript.

## Abstract

Autosomal recessive conditions are often associated with homozygous mutations showing common ancestral origins and are frequently linked to consanguinity. However, an increasing number of compound heterozygotes are found in diverse, admixed populations. Oculocutaneous albinism (OCA) is a recessive condition caused mainly by mutations in the *TYR* and *OCA2* genes involved in skin pigmentation. We previously screened the *TYR* and *OCA2* genes in Colombian OCA families, identifying both known and novel mutations. Affected family members were found to be either homozygous or compound heterozygous for these gene mutations. Compound heterozygosity, where two different recessive alleles inherited from each parent lead to the expression of an autosomal recessive trait, poses a challenge in genetic diagnosis. Estimating the ancestry of these disease-associated variants, in conjunction with understanding the colonization history of admixed populations, can enhance the precision of association mapping in genetic studies. The aim of this study was to determine the ancestral origins of *TYR* and *OCA2* mutations for OCA patients from two populations located in the Andes region of Colombia–Altiplano Cundiboyacense and Marinilla-Santuario. Comparison of OCA patients, and their unaffected relatives, with global reference populations showed a pattern of European and Native American admixture, with little African ancestry, for these two populations. Mutation-bearing *TYR* and *OCA2* haplotypes from Colombian OCA patients were compared against haplotypes from Spanish, Native American, and Sephardic Jewish reference populations to infer their ancestral origins. For 12 OCA1 patients from the Altiplano Cundiboyacense region, 21 out of 24 mutated *TYR* haplotypes show Spanish origins, two show Native American origins, and one shows a Sephardic Jewish origin. The two Native American *TYR* haplotypes, and the single Sephardic Jewish haplotype, are all found in compound heterozygote patients, paired with the predominant Spanish *TYR* haplotype G47D. OCA in these three patients is a result of genetic admixture that brought together disease-causing mutations with distinct ancestral origins. Both OCA2 patients from the Marinilla-Santuario region show homozygous OCA2 mutations with a Spanish origin. These findings underscore the complexity of the genetic architecture

**Funding:** This work was supported by the Facultad de Ciencias at Universidad de los Andes through the 2018-1 Research Project Funding Call and the 2023 Call (INV-2023-162-2849) (awarded to MCL). Additional funding was provided by the Georgia Institute of Technology (RF383) (awarded to KIJ). The funders had no role in study design, data collection and analysis, decision to publish, or preparation of the manuscript.

**Competing interests:** NO authors have competing interests

of Mendelian disease in admixed American populations, with both consanguinity and admixture contributing to the risk of autosomal recessive OCA in Colombia.

## Introduction

Colombia has a highly admixed population with African, European, and Native American ancestry components [1–4]. Prior to the arrival of Spanish conquistadors in 1525, Colombia was populated by various Native American groups, including the Muisca who inhabited the central Andean highland region (corresponding to the modern departments of Boyacá and Cundinamarca) and the Quimbaya who inhabited the Cauca river valley to the west and south (corresponding to the modern departments of Caldas, Quindío, and Risaralda) [5]. The early period of Spanish colonization resulted in European and Native American admixture, which was sex-biased with predominantly male European and female Native American components, and later the transatlantic slave trade introduced an African ancestry component [1, 4, 6–9]. A smaller number of Conversos–Sephardic Jews who converted to Catholicism owing to persecution in Spain and Portugal–also immigrated to Colombia in the 16th and 17th centuries, and a subsequent wave of Jewish immigration to Colombia occurred during and after World War II [10–12].

The degree of African, European, and Native American admixture varies among the five natural regions of Colombia: Amazonian, Andean, Caribbean, Orinoquian, and Pacific [13]. This study is focused on the Andean region, which is characterized by relatively high levels of European and Native American ancestry compared to low levels of African ancestry [11, 14, 15]. Discernable levels of Converso genetic ancestry were also recently discovered in Colombian populations sampled from this region [11]. To date, the vast majority of clinical genomics research has been conducted on European ancestry participant cohorts, with genomes from Latin America being particularly underrepresented [16–18]. For example, as of May 2024, the genome-wide association study (GWAS) diversity monitor shows that Hispanic or Latin American individuals make up a mere 0.38% of GWAS study participants [19]. The Eurocentric bias in genomics research limits the global reach of precision medicine and obscures the impact of human migration and evolution on genetic disease. We are interested in the relationship between genetic ancestry, admixture, and the presence of mutations that cause oculocutaneous albinism (OCA) in Colombian populations from the Andean region. The populations studied here live in the Andean areas of Altiplano Cundiboyacense and Antioquia, which are located in the eastern (cordillera oriental) and central (cordillera central) Andes mountain ranges, respectively, separated by the Magdalena River (Fig 1A).

There are two subtypes of OCA; OCA1 (type I) is caused by mutations in the tyrosinase gene (*TYR*), and OCA2 (type II) is caused by mutations in the OCA2 melanosomal transmembrane protein gene (*OCA2*) [20, 21]. OCA1 is characterized by extremely pale skin, white hair, and light-colored irises, whereas OCA2 is typically less severe than OCA1 and characterized by creamy white colored-skin and hair that can be light yellow, blond, or light brown. We previously described seven different mutations in the *TYR* gene responsible for OCA1 in individuals from Altiplano Cundiboyacense and a single *OCA2* mutation responsible for OCA2 in individuals from towns of Marinilla and Santuario in Antioquia [22, 23]. The most common *TYR* mutation observed in Altiplano Cundiboyacense was G47D (dbSNP rs61753180) found in 82% of OCA1 cases. The G47D mutation was found as homozygous or as compound heterozygous together with either the 1379delTT, 580delA, or S184X mutations of the same *TYR*

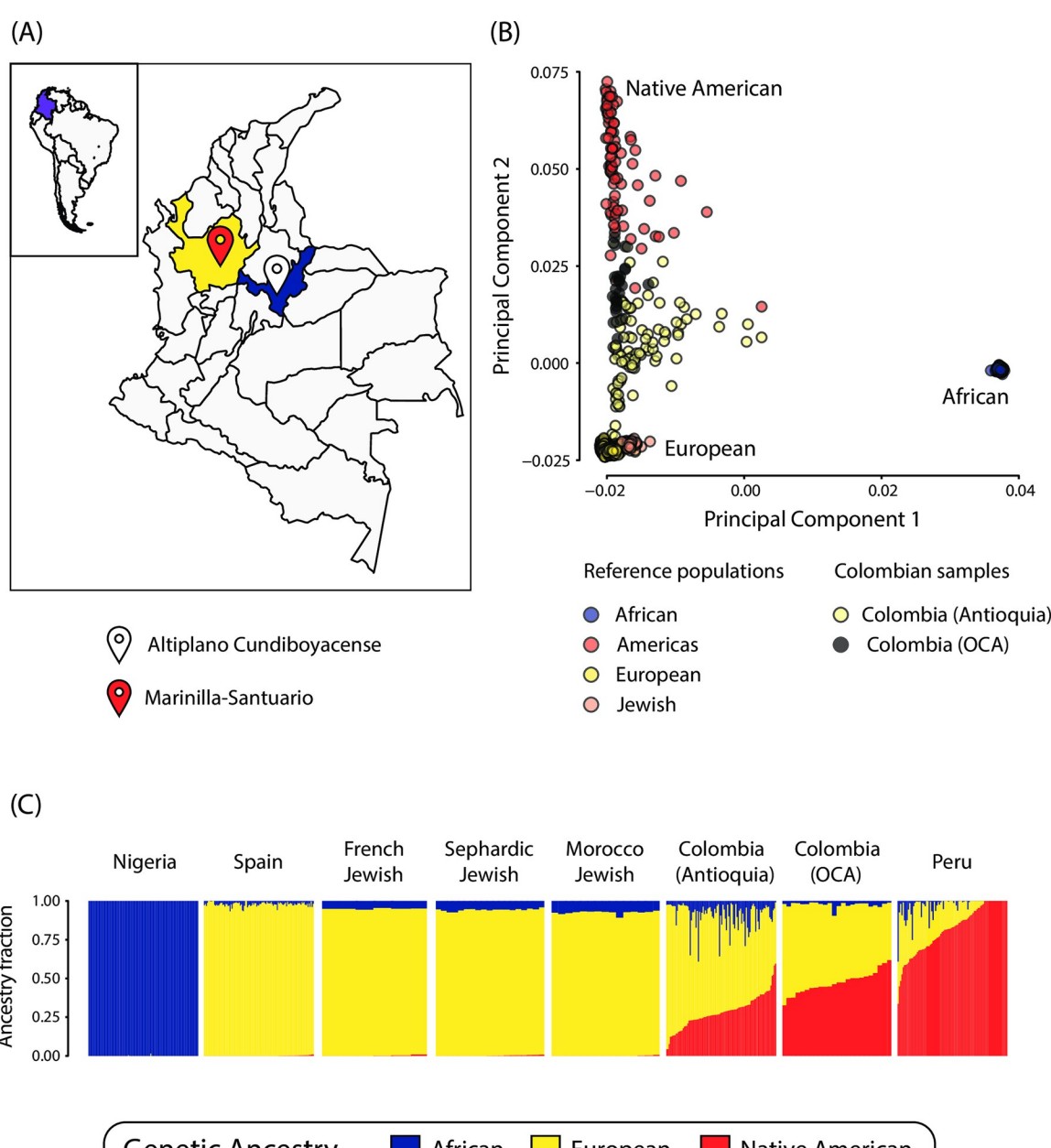

**Fig 1. Genetic ancestry and admixture in Altiplano Cundiboyacense and Marinilla-Santuario.** (A) Locations of Altiplano Cundiboyacense (white) and Marinilla-Santuario (red) in the Colombian administrative departments of Boyacá (blue), and Antioquia (yellow). The map file used to generate the image is freely distributed for MapSVG and is licensed under the Creative Commons Attribution 4.0 International Public License (URLs: https://mapsvg.com/maps/world and https://mapsvg.com/maps/colombia). (B) Principal component analysis (PCA) projection of genomic distances between Colombian and global reference populations. (C) ADMIXTURE plot showing African (blue), European (yellow), and Native American (red) ancestry fractions for individuals from Colombian and global reference populations.

gene. A787T (dbSNP: rs2063883437) was the only *OCA2* mutation found in the Antioquia towns of Marinilla and Santuario, where high numbers of individuals with OCA2 reside.

Given the ancestral diversity of Colombian populations, together with our previous observations of *TYR* compound heterozygotes in Altiplano Cundiboyacense, we hypothesized that

admixture resulting from the Spanish colonization of Colombia may have led to multiple *OCA* mutations of distinct ancestries segregating in individuals from the Andean region. We refer to this phenomenon as the admixture-derived compound heterozygote hypothesis [24, 25]. In support of this idea, the G47D mutation was suggested to have a Sephardic Jewish origin based on its high prevalence among Moroccan Jews living in Israel [26]. This same mutation was found in OCA patients living in the Canary Islands and Puerto Rico, suggesting the possibility of a common origin linked to the Sephardic Jewish diaspora [26, 27]. Given the potential admixture-derived compound heterozygote observed in families from the Andean regions, along with the reported origin of *OCA* mutations and Colombia's history of multiple waves of immigration from various ancestral groups (including Spanish and Jewish populations), we decided to perform haplotype inference analysis for each of the previously reported *OCA*-related mutations. Through this analysis, we aim to provide valuable insights into the genetic complexities of admixed populations, including the development of autosomal recessive conditions due to genetic admixture and the influence of inbreeding and outbreeding depression on mating patterns among unrelated individuals.

## Methods

### Study cohort and ethics

The study cohort consists of 19 individuals who were enrolled in March 2017 and sampled for a prior study on the genetic etiology of OCA in Colombia [22, 23]. Our initial study included 36 individuals with OCA from 23 independent families across various Colombian regions, recruited through the Albino Organization 'Fundación Contraste–Albinos por Colombia [22, 23]. The current study includes nine individuals from a single family in Altiplano Cundiboya-cense, including four OCA1-affected and five unaffected individuals. Eight additional unrelated OCA1-affected individuals were sampled from Altiplano Cundiboyacense and nearby departments. Two unrelated OCA2-affected individuals were sampled from in or nearby the Marinilla-Santuario region in Antioquia (Fig 1A). The same pedigree data and nomenclature reported by our group [22, 23] are preserved in this study.

This study conforms to the Helsinki ethical principles for medical research involving human subjects. The Albinism project was evaluated and approved by the Universidad de los Andes Research Ethics Committee and determined to comply with all scientific, technical, and administrative standards for health research established by the 1993 Resolution #008430 of the Colombian Ministry of Health. The project was classified as minimal risk by the ethics committee. Participant recruitment was done in collaboration with local community representatives, and vulnerable individuals were excluded from participation. All participants or their legal guardians signed informed consent, indicating their understanding of the genetic testing procedure, the benefits, limits, and possible consequences of the test, and granting permission for basic genetic research to be conducted on their samples. Access to participant samples was restricted to the project principal investigator (MCL).

### Genetic variant analysis

DNA samples were accessed in March 2017. *TYR* and *OCA2* mutations from participant samples were examined using Sanger sequencing as described in previous studies on these individuals [22, 23]. The identities of pathogenic mutations were confirmed based on disease diagnosis, previous OCA literature, the ClinVar and OMIM databases, and patterns of inheritance (homozygous or compound heterozygous).

These individuals were genotyped with the Infinium OmniExpress-24 BeadChip (Illumina) using the Macrogen service (Seoul, Republic of Korea). The 24 Omni Express Chip evaluates

genotypes at ~720,000 sites genome-wide. Genome-wide genotype data from the Colombian cohort were merged and harmonized with genomic variant data from global reference samples to facilitate genetic ancestry inference. The ancestry of TYR haplotypes was evaluated using a 318kb region, +/- 100kb from the gene transcription start and end sites (chr11:89,077,875–89,395,759), consisting of 47 SNPs. African (YRI, n = 108), European (IBS, n = 107) and Native American (PEL, n = 85) ancestry reference samples, along with a Colombian population from Medellín (CLM, n = 94), were from the 1000 Genomes Project [28]. Genomic variant data from Sephardic Jewish populations (JWS, n = 121) were taken from a study on Jewish genetic ancestry [29]. Genomic variant merging and harmonization were done using PLINK version 1.9 and custom scripts [30]. A variant missingness filter of 5% was used together with a minor allele frequency filter of 1%. Linkage disequilibrium pruning was performed using the "-indep" command in PLINK 1.9 with a window size of 50 kb, a step size of 5 variants, and a variant inflation factor threshold of 2. Phasing was performed using ShapeIT version 2.r837 [31].

## Genetic ancestry inference

The merged and harmonized genomic variant dataset was used for genome-wide and local (i.e. haplotype around the TYR and OCA genes) ancestry inference. Principal component analysis was performed to examine ancestry divergence via comparison with genomic variant data from African, European, and Native American reference populations by using "-pca" option in PLINK version 1.9. Genome-wide ancestry inference was performed using the program ADMIXTURE version 1.30 [32]. Local ancestry inference, i.e. the assignment of ancestral origins for individual haplotypes across the genome, was performed using RFMix version 2 [33]. TYR and OCA2 haplotype regions were operationally defined as ±50kb from the start and stop site of each gene, and haplotype ancestral origins were taken from RFMix output. Ancestry origins were only considered when the RFMix ancestral certainty was at least 95%. TYR haplotype sequences were extracted and analyzed using MEGA version X to visualize haplotype relationships [34]. Pairwise identity-by-state distances were computed for all pairs of TYR haplotypes, and the resulting distance matrix was used to reconstruct a neighbor-joining phylogeny [35].

## Results

### Genetic ancestry and admixture of Colombian OCA patients

Principal component analysis (PCA) shows that the Colombian OCA patients studied here group closely together and fall between the European and Native American reference populations along principal component 2 (PC2; Fig 1B). The Colombian OCA patients are more tightly grouped in PCA spaced compared to a previously studied Colombian population from Medellín in Antioquia, which extends further towards the European reference population along PC2 and slightly towards the African reference population along PC1. Genetic admixture analysis, using the same global reference populations, confirmed that the Colombian OCA patients have a more even two-way European and Native American admixture pattern compared to the previously studied Colombian population from Medellín (Antioquia), which shows greater overall variation in admixture, with relatively more European and African ancestry and less Native American ancestry (Fig 1C).

### Ancestral origins of *TYR and OCA2* mutations

Next, TYR and OCA2 haplotypes for Colombian OCA patients were compared to orthologous haplotypes from Spanish, Native American, and Sephardic Jewish reference populations to

infer their ancestral origins. Pairwise identity-by-state distances were calculated between haplotypes and used to reconstruct haplotype phylogenies as illustrated for a subset of *TYR* haplotypes in Fig 2. The *TYR* G47D haplotype was inferred to have a Spanish origin owing to its sequence identity with Spanish-origin haplotypes sampled from Spanish (IBS) and Peruvian (PEL) populations. The *TYR* 1379delTT haplotype was inferred to have Native American origins by its clustering within a clade of Native American origin haplotypes sampled from a Peruvian population. The Spanish-origin G47D haplotype is found in 21 out of the 24 mutated TYR haplotypes among the 12 OCA1 patients evaluated here (Table 1). Two of the four affected family members were confirmed to be G47D homozygous, one was found to be G47D-1379delTT compound heterozygous, and one was characterized as G47D with a second unknown mutation. The individual with the second unknown mutated *TYR* haplotype is likely to be G47D homozygous based on the Spanish origin of that second haplotype. Three out of eight of the unrelated OCA1 patients were G47D homozygous, and three were found to be G47D-D42N compound heterozygous. The D42N haplotype also shows a Spanish origin.

The remaining two unrelated OCA1 individuals were compound heterozygotes, where each haplotype had a distinct ancestral origin. One of these individuals has a combination of the predominant Spanish-origin G47D haplotype and a second Native American-origin 580delA mutated haplotype, and the other individual had a combination of the predominant Spanish-origin G47D haplotype and a Sephardic Jewish-origin S184X haplotype. Both of the unrelated OCA2 patients evaluated here were homozygous for the Spanish-origin A787T OCA2 haplotype.

The pedigree in Fig 3 shows the inheritance patterns and ancestral origins for OCA1 causing *TYR* mutations, and the haplotypes on which they reside, in a single Colombian family. The inheritance patterns and haplotype origins for three out of the four affected individuals (V-26, V-27, and V-29) can be inferred based on the pedigree, whereas the fourth affected individual (VI-1) has an unknown inheritance and one uncharacterized mutated haplotype. Affected individuals V-26 and V-27 are siblings who inherited Spanish-origin G47D haplotypes from each of their unaffected (heterozygous) parents. Affected individual VI-29 is a compound heterozygote with a paternally inherited Spanish origin G47D haplotype and a maternally inherited Native American origin 1379delTT haplotype.

## Discussion

Our study on the genetic etiology of OCA in Colombia highlights the valuable insights that can be obtained from clinical genomics studies of Latin American populations with significant genetic admixture. Colombia serves as an ideal setting for studies like this one aimed at exploring the relationship between ancestry, admixture, and genetic determinants of health. The Colombian population has African, European, and Native American ancestry components, with distinct admixture patterns seen for different regions in the country [1–4, 6–9, 11, 14, 15]. African genetic ancestry is highest in the Pacific and Caribbean regions, whereas Native American ancestry is highest in the Amazonian and Orinoquian regions. The study population in the Andean region, mainly has European and Native American ancestry, and as the data revealed, a notable Converso ancestry component (Fig 1) [11].

OCA, an autosomal recessive condition, exhibits variable prevalence rates globally, ranging from 1 in 1,755 to 1 in 15,000 individuals [36]. Our group previously reported an elevated number of OCA cases in small towns within the Colombian Andean region [22, 23]. Furthermore, affected individuals carry mutations with unknown global frequencies in the gnomAD database [37] and exhibit compound heterozygosity. Here, examining relationship between genetic ancestry, admixture, and pattern of OCA mutation in Colombia, we found OCA1

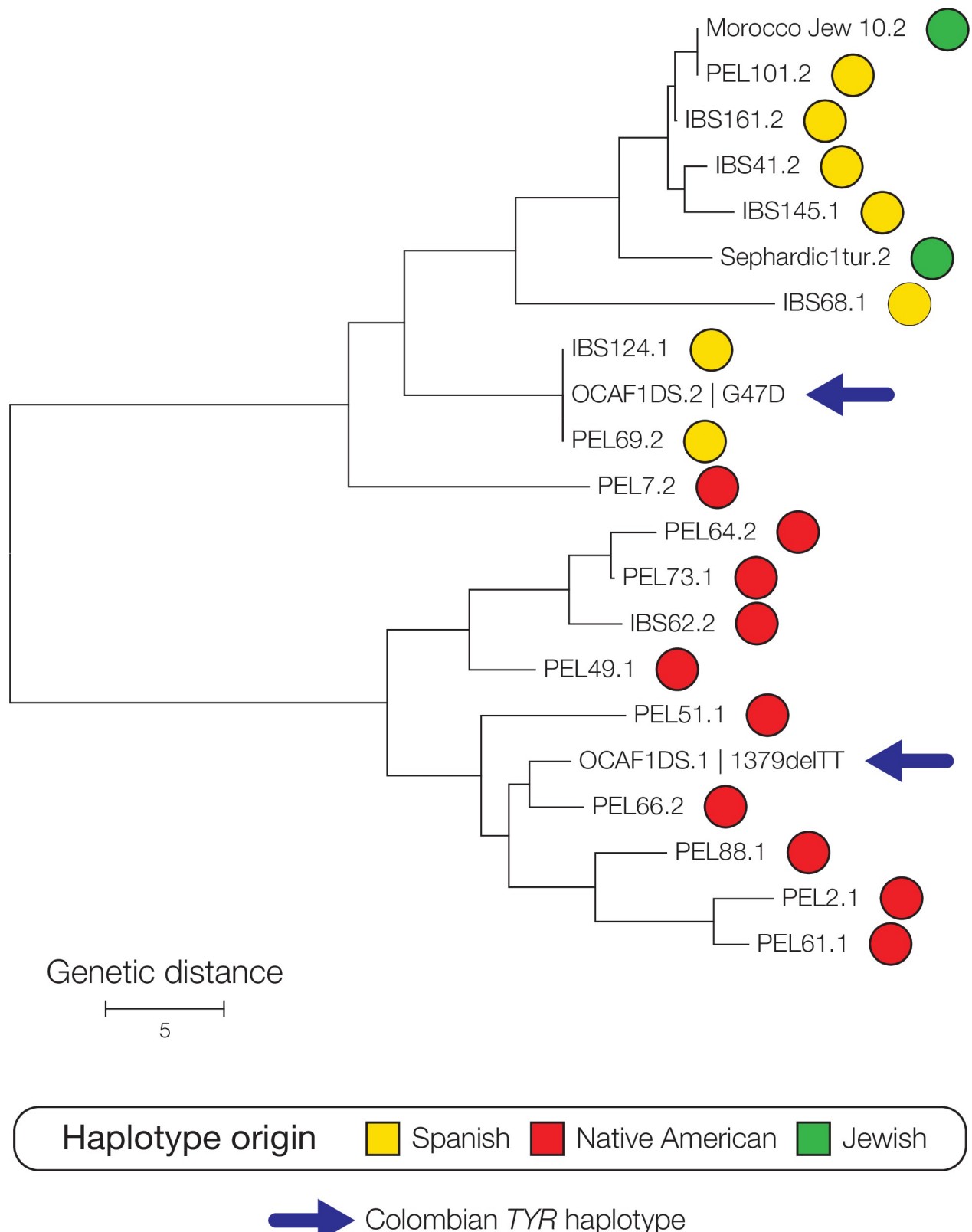

**Fig 2. Ancestry inference for the oculocutaneous albinism (OCA) causing TYR gene haplotypes G47D and 1379delTT.** Neighbor-joining phylogeny based on pairwise genetic distances between TYR gene haplotypes. The Colombian G47D and 1379delTT haplotypes are shown along

with their most closely related reference haplotypes. Reference haplotypes are color-coded according to their ancestral (population) origins as shown.

patients from our Altiplano Cundiboyacense study population show *TYR* mutations with ancestral origins from all three of these groups (Fig 2 and Table 1). The Spanish-origin G47D mutation is the most common, found in 21 out of 24 OCA1-linked haplotypes studied here, and there are two distinct Native American-origin *TYR* mutations and a single Sephardic Jewish-origin *TYR* mutation. All three of the non-Spanish-origin *TYR* mutations are found as compound heterozygotes in combination with the predominant Spanish-origin G47D mutation. The two OCA2 patients from the Marinilla-Santuario region show a single Spanish-origin OCA2 mutation A787T.

It was previously proposed that the G47D haplotype had a Sephardic Jewish origin [27], but here we demonstrate a Spanish origin for this haplotype (Fig 2 and Table 1). Considered together with the presence of G47D mutations in the Canary Islands and Puerto Rico [26], our finding is consistent with G47D as a founder mutation that was established in Colombia by Spanish migration and colonization in the New World. The G47D founder mutation in Altiplano Cundiboyacense could be linked to the small founding population of Spanish colonizers who settled in the town of Ciénega, in an area previously inhabited by the Muisca, located in

**Table 1. *TYR* and *OCA2* mutations and haplotype origins.**

| Pedigree[a] | Gene | Genotype[b] | Region | Haplotype Origin | Phenotype[c] |
|---|---|---|---|---|---|
| **Family with OCA1 (*TYR* gene)** | | | | | |
| VI-29 | *TYR* | G47D \| 1379delTT | Altiplano Cundiboyacense | Spanish\| Native American | Complete albinism |
| VI-28 | *TYR* | WT \| WT | Altiplano Cundiboyacense | Spanish\| Native American | Without albinism |
| VI-30 | *TYR* | WT \| 1379delTT | Altiplano Cundiboyacense | Native American \| Native American | Without albinism |
| VI-1 | *TYR* | Unknown \| G47D | Altiplano Cundiboyacense | Spanish \| Spanish | Complete albinism |
| V-19 | *TYR* | G47D \| WT | Altiplano Cundiboyacense | Spanish\| Native American | Without albinism |
| V-20 | *TYR* | WT \| 1379delTT | Altiplano Cundiboyacense | Spanish\| Native American | Without albinism |
| V-26 | *TYR* | G47D \| G47D | Altiplano Cundiboyacense | Spanish \| Spanish | Complete albinism |
| V-27 | *TYR* | G47D \| G47D | Altiplano Cundiboyacense | Spanish \| Spanish | Complete albinism |
| IV-11 | *TYR* | WT \| G47D | Altiplano Cundiboyacense | Native American \| Spanish | Without albinism |
| **Unrelated individuals with OCA1 (*TYR* gene)** | | | | | |
| | *TYR* | G47D \| G47D | Altiplano Cundiboyacense | Spanish \| Spanish | Complete albinism |
| | *TYR* | G47D \| G47D | Altiplano Cundiboyacense | Spanish \| Spanish | Complete albinism |
| | *TYR* | G47D \| G47D | Altiplano Cundiboyacense | Spanish \| Spanish | Complete albinism |
| | *TYR* | G47D \| D42N | Altiplano Cundiboyacense | Spanish \| Spanish | Complete albinism |
| | *TYR* | G47D \| D42N | Altiplano Cundiboyacense | Spanish \| Spanish | Complete albinism |
| | *TYR* | G47D \| D42N | Meta | Spanish \| Spanish | Complete albinism |
| | *TYR* | G47D \| 580delA | Nariño | Spanish\| Native American | Complete albinism |
| | *TYR* | G47D \| S184X | Tolima | Spanish\| Sephardic Jewish | Complete albinism |
| **Unrelated individuals with OCA2 (*OCA2* gene)** | | | | | |
| | *OCA2* | A787T \| A787T | Marinilla-Santuario | Spanish \| Spanish | Partial albinism |
| | *OCA2* | A787T \| A787T | Caldas | Spanish \| Spanish | Partial albinism |

[a] Pedigree code for individual family members as shown in Fig 3.

[b] Amino acid changes are shown with the protein-level mutation code, with the affected amino acid (one-letter code), its position, and the mutated amino acid. DNA-level deletions are shown with the position and the deleted nucleotides. WT refers to wild-type sequences with no albinism-causing mutation.

[c] Complete albinism refers to the more severe OCA1 (type I) associated phenotype, and partial albinism refers to the milder OCA1 (type II) associated phenotype.

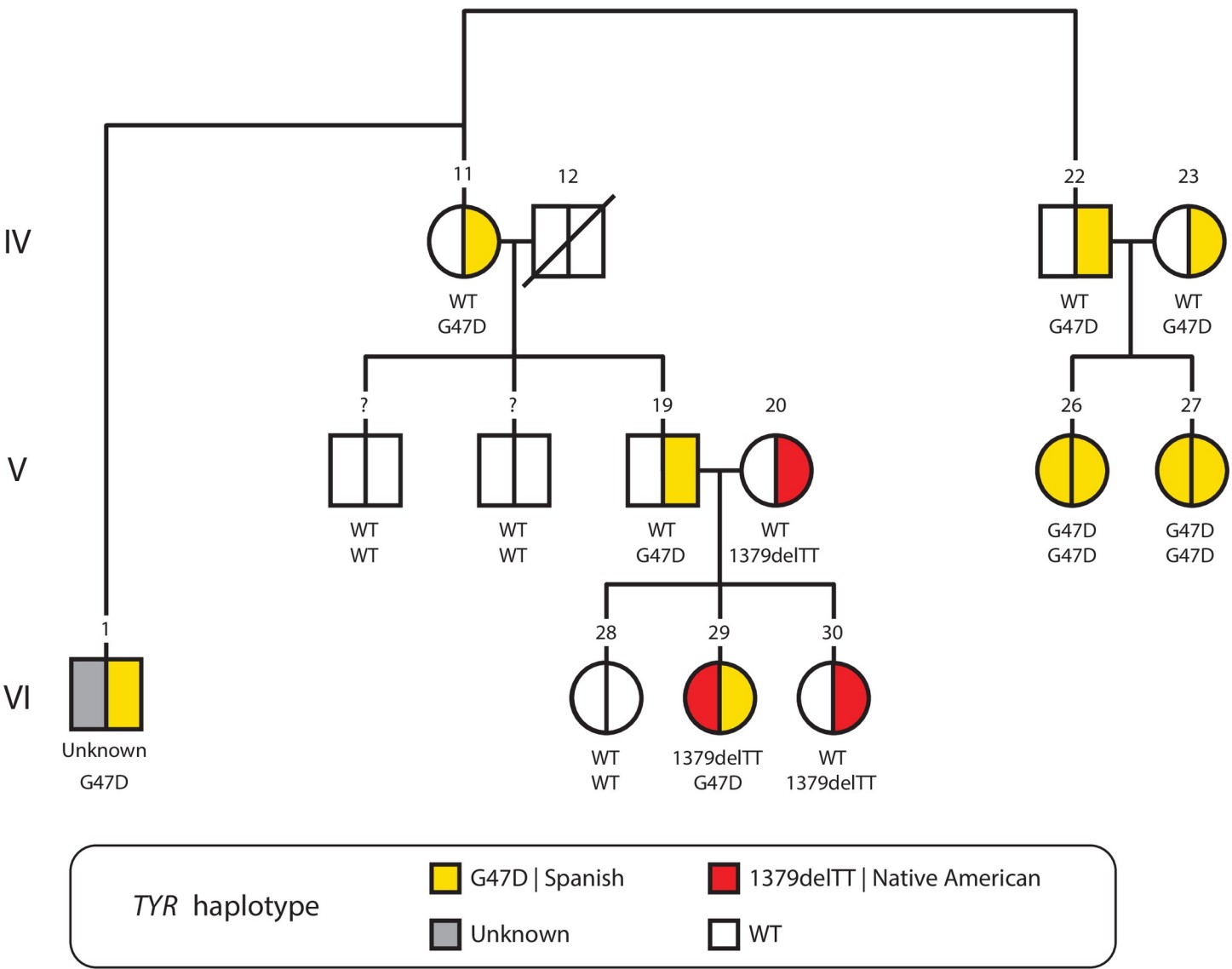

**Fig 3. Origin of oculocutaneous albinism (OCA) causing TYR gene haplotypes in a Colombian family.** Pedigree circles represent females, squares represent males, and the deceased individual is marked with a crossed bar. The pedigree shown here is part of a larger seven-generation pedigree for the extended family, with generations and individual codes indicated as in the original pedigree [23]. Wild-type (WT) TYR haplotypes are shown as white, whereas mutated OCA haplotypes are color-coded according to their ancestral origins: yellow for Spanish and red for Native American. Heterozygous carriers are half-shaded, and OCA-affected individuals are fully shaded. The specific identities of the two TYR haplotypes, WT or disease-causing, are indicated for each individual. The specific disease-causing mutation for one of the haplotypes in individual VI-1 is unknown and shaded gray here.

the modern department of Boyacá. It should be stressed that these two views on the origins of the G47D mutation may not be mutually exclusive. Sephardic Jews are a Jewish diaspora population from the Iberian Peninsula, who were eventually exiled from Spain following the Alhambra Decree of 1492 and fled to North Africa [38]. Thus, the presence of the G47D haplotype among Moroccan Jews in Israel could be attributed to the exile of Sephardic Jews from 15th-century Spain to Morocco followed by a subsequent expulsion from Morocco and migration to Palestine in the 19th century or the modern state of Israel in the 20th century.

The two OCA2 patients from Marinilla-Santuario evaluated here are both homozygous with Spanish-origin A787T mutations. This observation and the previous finding that the

Marinilla-Santuario region had the highest inbreeding coefficient in Colombia [39], is consistent with the idea that a founder event influenced variation in this region in particular. However, given the relatively low sample size of two Marinilla-Santuario samples, and the lack of pedigree linking these samples, this conclusion is not as well supported as the G47D founder mutation.

The pedigree shown in Fig 3 underscores the complex inheritance patterns for OCA1--linked TYR mutations in a single Colombian family. There are three OCA patients in this pedigree, for which both TYR mutations and haplotype origins are analyzed. Two of these individuals, who are siblings, are homozygous for the same G47D haplotype inherited from both parents, pointing to some degree of consanguinity in this family. This is consistent with a known role for consanguinity in many autosomal recessive disorders [40, 41], such as thalassemia, cystic fibrosis, and Tay-Sachs disease, and represents a kind of inbreeding depression where reduced fitness of individuals with genetically related parents is caused by recessive deleterious mutations that are identical by descent [42].

In contrast to the presumed OCA founder effect mutations seen for the *TYR* and *OCA2* genes, one of the Altiplano Cundiboyacense family members shown in Fig 3, and two of the eight unrelated OCA patients from Altiplano Cundiboyacense, also show evidence for compound heterozygous mutations with distinct ancestral origins. Two of these cases show Spanish-Native American haplotype combinations and one shows a Spanish-Sephardic Jewish haplotype combination (Table 1). These cases point to a role for genetic admixture in generating disease-causing compound heterozygotes, which could represent an example of outbreeding depression in humans [43, 44]. This diversity of haplotype combinations is consistent with the larger population of the Altiplano Cundiboyacense region, compared to Marinilla-Santuario, including the capital city of Bogotá, which has a large and ethnically diverse population of ~8 million inhabitants.

The interplay between deleterious variants, genetic admixture, and inbreeding has been documented in Antioquia—Colombia, particularly in relation to complex disorders like Alzheimer's disease, frontotemporal lobar degeneration and early-onset dementia. A case of early-onset Alzheimer's disease was identified in an individual who was a compound heterozygote, carrying both the African-origin Thr96Lys/Trp191*/Leu211Pro haplotype and the Native American c.469C>T (p.His157Tyr) variant in the *TREM2* gene [45]. Additionally, the c.140G>A (p.Arg47His) variant in the *TREM2* gene, associated with European ancestry, was found in both homozygous and heterozygous carriers [45]. These findings reinforce the significant role of genetic admixture in shaping the prevalence of inherited disorders.

In conclusion, we find evidence of both consanguinity-linked founder effect mutations, with Spanish ancestral origins, and admixture-generated compound heterozygotes, with hybrid Spanish, Native American, and Sephardic Jewish ancestral origins, among the Colombian OCA patients studied here. These results highlight the complex genetic architecture of autosomal recessive diseases in admixed American populations and underscore the kinds of insights that can be gained by studying disease genetics in ancestrally diverse populations.

## Acknowledgments

The authors wish to thank all the members of the OCA-affected families who took part in the study. They also wish to thank the Departamento Administrativo de Ciencia, Tecnología e Innovación–Colciencias (Call 617–2013) and the Fondo de Apoyo a Doctorados de la Universidad de los Andes (2017) for their support of Wilmer Cardenas's PhD program, as well as Fundación Ceiba for funding Diana Núñez's doctoral studies.

## Author Contributions

**Conceptualization:** Wilmer A. Cárdenas, María Claudia Lattig.

**Formal analysis:** Wilmer A. Cárdenas.

**Funding acquisition:** María Claudia Lattig.

**Methodology:** Wilmer A. Cárdenas, Andrew B. Conley, Shashwat Deepali Nagar, I. King Jordan.

**Supervision:** María Claudia Lattig.

**Writing – original draft:** Wilmer A. Cárdenas.

**Writing – review & editing:** Wilmer A. Cárdenas, Diana L. Núñez-Ríos, María Claudia Lattig.

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
