## [Decision Letter · Decision Letter 0]

20 Mar 2024

PONE-D-23-39938Ancestral origins of TYR and OCA2 gene mutations in Oculocutaneous albinism from two admixed populations in ColombiaPLOS ONE

Dear Dr. Nunez-Rios,

Thank you for submitting your manuscript to PLOS ONE. After careful consideration, we feel that it has merit but does not fully meet PLOS ONE’s publication criteria as it currently stands. Therefore, we invite you to submit a revised version of the manuscript that addresses the points raised during the review process.

The study represents a small but curious study of genes influencing a Mendelian trait in admixed human populations. We had a hard time securing reviewers and the editor eventually took on the role of reviewing. We apologize for the delay. The work is of sufficient quality for PLOS one, but the MS lacks some polishing of narrative and the writing itself.

See the very excellent points by the reviewer, and the following from the editor.

The three major points are

The results section lacks context and focus. Specifically, the second paragraph (Line 119.) dives straight to specifics and plenty of un-defined abberviations are used (IBS, NAM). Restructure this part, open with a general results statement that captures the details the follow. “First we analyzed a familial case, and then studied unrelated individuals…” Use the structure. We wanted to know … To addrss that we did … The data revealed … (rephrase as needed). See also minor points belowThe population genetic methods are lacking in detail. Starting Line 248. This includes how many samples analyzed? How many SNPs on this chip? How many polymorphic in your sample? How many markers / invidividuals were used in each analyses? Same dataset in all or subsets in some? Which version of MEGA? How wide a region around OCA/TYR?The language needs considerable work. See numerous smaller pointers from the editor below, but you are strongly encouraged to scrutinize the manuscript for English grammar errors because many must have eluded me. Getting a 3^rd^ party or native english friend to read may help. ==============================

We look forward to receiving your revised manuscript.

Kind regards,

Arnar Palsson, Ph.D.

Academic Editor

PLOS ONE

2. We note that your Data Availability Statement is currently as follows: [All relevant data are within the manuscript and its Supporting Information files]

Additional Editor Comments:

The study represents a small but curious study of genes influencing a Mendelian trait in admixed human populations. We had a hard time securing reviewers and the editor eventually took on the role of reviewing. We apologize for the delay. The work is of sufficient quality for PLOS one, but the MS lacks some polishing of narrative and the writing itself.

See the very excellent points by the reviewer, and the following from the editor.

The three major points are

1. The results section lacks context and focus. Specifically, the second paragraph (Line 119.) dives straight to specifics and plenty of un-defined abberviations are used (IBS, NAM). Restructure this part, open with a general results statement that captures the details the follow. “First we analyzed a familial case, and then studied unrelated individuals…” Use the structure. We wanted to know … To addrss that we did … The data revealed … (rephrase as needed). See also minor points below

2. The population genetic methods are lacking in detail. Starting Line 248. This includes how many samples analyzed? How many SNPs on this chip? How many polymorphic in your sample? How many markers / invidividuals were used in each analyses? Same dataset in all or subsets in some? Which version of MEGA? How wide a region around OCA/TYR?

3. The language needs considerable work. See numerous smaller pointers from the editor below, but you are strongly encouraged to scrutinize the manuscript for English grammar errors because many must have eluded me. Getting a 3rd party or native english friend to read may help.

Minor comments.

Italicize gene names in text.

Line 26.

Does prior knowledge lead to this conclusion “patients from two regions of Colombia that show distinct patterns of genetic ancestry“? Or is this a finding from this study? Could “with“ work instead?

Line 29

OCA1 (and OCA2 below) show up as undefined abbreviations. Are the trait subtypes are do they refer to the genetics?

Line 36.

Reword “region have homozygous… of spanish”

Line 46.

Not right choice of words “On the contrary,” In contrast??

Line 49

Drop “of receiving”

Line 56.

Reword “goal” is not the right term her.

Line 60

Reword sentence. “underserved” is not the right word here

Line 64

First statement is categorically wrong. Genetic disorders can be studied on other continents just fine in absence of this history. Open with something more general.

Line 71

Why roman numbers and referring to centuries? Use regular decades per century please.

Line 79.

Why a special interest in this region? Provide justificaiton, can be pragmatic.

Line 85

No need to say “In Colombia,”, that is pretty clear at this point.

Line 87

Can shorten, drop “both regions are shown in” and just retain “(Figure 1A)”

Line 94

Refer to numbers behind high numbers of “individuals with OCA2”.

Line 98

Reword entire sentence “In this regard, the” Could start with “for example…”

Line 100

Drop “One year later,”

Line 105

Reword the opening sentence of this paragraph. If you want a preamble, refer to the important part of genetics that needs resolving, before stating your aim (and then the local/genetic details of the study system). Also state the specific research questions more clearly. “We asked if/whether ….”

Line 112

Add “can” after outbreeding.

Line 115

Explain better the geography or separation of the two populations studied, that is relevant for this study! The opening as stands does not work. Maybe “we studied OCA in populations from two regions are in the Columbian Andes “ (highlight differences between regions??)

Line 116.

From what data were these graphs generated? What do the graphs show the reader? Subgroups and level of mixing in these two regions??? Cite sources of these data if not newly generated for this study.

Line 132

Reword “With regards to the A787T mutation in the OCA2 gene in Marinilla and Santuario, they all have the same IBS haplotype” perhaps, “All the carriers of the … have the same IBS…”

Also highlight that here you swithc to antoher gene, after all the prior section focusing on TYR.

Line 134.

Explain how these haplotypes were derived/estimated. Add details into the sections above. Am not sure “Suggested” is the best term for the table legend. Name or region alti… lowercase in table. Last column, change to Phenotype (albinism) and recode as Complete, None/absent (without doesn’t work).

Line 138

Reword sentence “using…distance …” Perhaps just “Distances from … revealed”

And “from our samples with the Jews” is arkane wording. Fix. What is the point of this sentence?

Line 145

“is strikingly evident in genetic studies,” Add a citation to back up this statement.

Line 149.

“but also indicates how these phenomena influence in the development of autosomal recessive conditions” Really, am not sure the data show this!

Line 151

“departments”?? regions/ provinces??

Line 157

“magnificent” is not needed. This is not literture.

Line 164.

Reword. “Geographic divergence can result in regional isolation, influences assortative” This sentence talks about geographic separation can lead to divergence, but “geographic divergence” is not a good term.

Line 166.

Again, drop the romans.

Line 167

“matings” instead of “mattings”

Line 170

Move citation up in the sentence (0.0040) and reword this last part. “shown by the A787T homozygous mutation in the OCA2 gene” Maybe “And the homoz… may reflect this level of inbreeding.

Line 173

Not exactly true, two and distinct concepts “Inbreeding is often referred as consanguinity” Reword. Or, is this part needed??

Line 175

“is characteristic” maybe “follows”?

Line 178

Plural “populations”

Line 181 and 183

Bad wording, sentence starts and ends with same wording. And in any rewrite, drop “phenomena”

Line 184

Don’t use “on the contrary”

Line 184.

State the main finding in clear terms at the beginning, before going into specifics.

Line 185.

Reword, some connecting words missing “is the largest city harboring individuals”

Line 190

Reword. Have no idea what “may be increase product of the breeding of distant populations as occur in admixed populations” means.

Line 200.

Reword sentence “effect in this high admixed region.” And should this not also affect other genetic variatns causing recessive diseses?

Line 201

Drop “to report potential effects of inbreeding and outbreeding in the admixed Andean region of Colombia”

Line 208.

Italic de novo

Line 224

Not good wording “with the Native American Human.” What is the acceptable wording in Columbia when referring to the aboriginies in the region?? Perhaps you want to do better, and acknowledge the hardship they have suffered at and since the colonization?

Line 231

“For the present study samples were accessed in March 2017.” What does this mean? Accessed from what?

Line 232.

“Ethical considerations were strictly observed, and” this has to be fleshed out, also in the ethics section. With reference to licences/permits for study.

Line 238

Plural “are homozygotes”

Line 238

Reword, something like

“The pedigree (Figure 2) was modifed, relative to earlier account of …”

Line 351

Which data used in this figure were newly generated for this study? What do the Columbian sub groups indicate (OCA and Antio…).

Line 367

Is this pedigree showing TWO genes? The coding of TYR and OCA is confusing.

According to legend “TYR haplotypes are shown as white, whereas mutated OCA haplotypes are color coded“ Im quite confused.

Explain in legend which gene does the variants status G47D refer to?

Figure 3.

Green and red colours are not friendly for the genetic carriers of colour blind genes. Please change colour palette.

Reviewers' comments:

Reviewer's Responses to Questions

**Comments to the Author**

1. Is the manuscript technically sound, and do the data support the conclusions?

Reviewer #1: No

2. Has the statistical analysis been performed appropriately and rigorously? 

Reviewer #1: No

3. Have the authors made all data underlying the findings in their manuscript fully available?

Reviewer #1: Yes

4. Is the manuscript presented in an intelligible fashion and written in standard English?

Reviewer #1: No

5. Review Comments to the Author

Reviewer #1: Ancestral origins of TYR and OCA2 gene mutations in Oculocutaneous albinism 2 from two admixed populations in Colombia.

The manuscript needs to be proof read with careful consideration to English language, grammar and sentence construction.

Introduction –

1. The Research study looks at the ancestral origins of 2 genes, mutations in which play an important role in manifestation of Oculocutaneous Albinism 2 (OCA2) in two admixed populations of Columbia.

2. The historical context about the region helps the reader understand the history of this region.

3. It is unclear from the present text as to why authors are interested in investigating mutations of TYR, and OCA2 genes. The research question needs to be better outlined. Elaborating this will explain the need to look at ancestral origins of OCA mutations.

4. This study builds on previous research done by the team, however inclusion of this data (in form of tables) needs to be incorporated and background work needs to be elaborated to build connections to present study (Page 5; 85-94; 106).

5. What are inbreeding estimates of the study populations? Authors should provide relevant data to support the statement that “By examining population migration in the Andes regions and ancestral origins of the described mutations we report how both inbreeding and outbreeding increase the risk to develop autosomal recessive traits.” (Page 6; 111-113)

Results –

6. Page 6; 116-118 – PCA analysis and global ancestry analysis should be explained for study populations.

7. The methods focus on haplotype comparisons using pedigree. The choice and justification of this methodological approach should be highlighted.

8. Results to be sectioned based on analysis. Detailed results to be explained for the pedigree and transmission of OCA mutations.

9. Role of inbreeding is not measured nor indicated in results as stated in Introduction

Discussion –

10. Page 8; 147-150 - This study is not investigating the complex demographic history of region but is trying to infer patterns of 2 gene mutations in the context of the complex demographic history of Columbia.

11. A large part of the discussion focuses on history and ancestry of these populations and region. While this is essential, authors should focus on genes under study and discuss their results in context of prior work and population history of the region.

12. Page 10; 181-183 – is there evidence of homozygosity due to inbreeding for other loci from this region to support this statement?

13. Page 10; 184-186 – Authors should provide reference of estimates of genetic diversity of Bogotá, the capital of Colombia.

14. Author’s proposal that compound heterozygotes found in individuals who carry mutations derived from varied ancestral sources can be due to outbreeding needs support from other population genetics studies. It may be useful to use population genetic models to explain how inbreeding and outbreeding affect compound heterozygotes.

Methodology –

15. Samples size is too limited to perform robust statistical analysis.

16. Methodological choice of taking pedigree data needs to be justified.

17. Authors should include a brief description of study participants and their recruitment here for clarity and not just cite the previous study.

18. How was the recruitment done for non-related individuals and what were ethical considerations while recruiting them.

6. PLOS authors have the option to publish the peer review history of their article (what does this mean?). If published, this will include your full peer review and any attached files.

Reviewer #1: **Yes: **Manjari Jonnalagadda, PhD

---

## [Author Response · Author response to Decision Letter 0]

24 Jun 2024

Editor comments and author responses

Editor summary. The study represents a small but curious study of genes influencing a Mendelian trait in admixed human populations. We had a hard time securing reviewers and the editor eventually took on the role of reviewing. We apologize for the delay. The work is of sufficient quality for PLOS one, but the MS lacks some polishing of narrative and the writing itself.

Author response to editor summary. We were pleased to read that editor and reviewer found our manuscript to be of interest and worthy of publication in PLOS ONE pending substantial improvements in the writing and narrative. We are particularly grateful that the editor took on the role of reviewing given the difficulty in securing reviewers. We had one of the native English-speaking coauthors (IKJ) completely rewrite the manuscript to improve the writing, focus, and narrative arc. We detail the changes made in response to the editor and reviewer comments below, indicating how and where we changed the manuscript in response to each point. 

See the very excellent points by the reviewer, and the following from the editor.

The three major points are

Editor comment # 1. The results section lacks context and focus. Specifically, the second paragraph (Line 119.) dives straight to specifics and plenty of un-defined abberviations are used (IBS, NAM). Restructure this part, open with a general results statement that captures the details the follow. “First we analyzed a familial case, and then studied unrelated individuals…” Use the structure. We wanted to know … To address that we did … The data revealed … (rephrase as needed). See also minor points below

Author response to editor comment #1. We have revised the entire Results section as suggested to improve context and focus. We are sure to define all abbreviations when they first appear in the manuscript, but we have largely tried to avoid unnecessary abbreviations throughout (eg see revised Table 1). We appreciate the suggestions for how to structure this part, and we have tried to follow them throughout (see revised Results section).

Editor comment #2. The population genetic methods are lacking in detail. Starting Line 248. This includes how many samples analyzed? How many SNPs on this chip? How many polymorphic in your sample? How many markers / invidividuals were used in each analyses? Same dataset in all or subsets in some? Which version of MEGA? How wide a region around OCA/TYR?

Author response to editor comment #2. We have revised the entire Methods section as suggested to provide all of the key missing details as prompted by the editor (see revised Methods section with new subsections on Genetic variant analysis and Genetic ancestry inference). 

Editor comment #3. The language needs considerable work. See numerous smaller pointers from the editor below, but you are strongly encouraged to scrutinize the manuscript for English grammar errors because many must have eluded me. Getting a 3rd party or native english friend to read may help.

Author response to editor comment #3. We are very grateful that the editor took the time and effort to provide so many suggestions for how to improve the grammar. As suggested, we had one of the native English-speaking coauthors completely rewrite the manuscript to improve the writing, focus, and narrative arc. Please note that instead of trying to improve the wording by addressing each of the editor’s very helpful grammar suggestions directly, we revised the entire manuscript. (Thus, we do not provide point-by-point responses to those comments). 

Reviewer #1 comments and author responses

Reviewer #1 summary. Ancestral origins of TYR and OCA2 gene mutations in Oculocutaneous albinism 2 from two admixed populations in Colombia. The manuscript needs to be proof read with careful consideration to English language, grammar and sentence construction.

Author response to editor summary. As suggested by the reviewer and editor, we had one of the native English-speaking coauthors completely rewrite the manuscript with careful consideration to the English language, grammar, and sentence construction. We greatly appreciate the comprehensive and thoughtful comments provided by the reviewer. We have addressed all of the reviewer’s comments and revised our manuscript accordingly. Below, we indicated how and where we revised the manuscript in response to each comment.

Introduction

Reviewer #1 comment #1. The Research study looks at the ancestral origins of 2 genes, mutations in which play an important role in manifestation of Oculocutaneous Albinism 2 (OCA2) in two admixed populations of Columbia.

Reviewer #1 comment #2. The historical context about the region helps the reader understand the history of this region.

Author response to reviewer #1 comments #1-2. We are glad that the reviewer found our study on the genetics of Albinism in two admixed populations of Colombia to be of interest and appreciate the recognition that the historical context is helpful to the readers.

Reviewer #1 comment #3. It is unclear from the present text as to why authors are interested in investigating mutations of TYR, and OCA2 genes. The research question needs to be better outlined. Elaborating this will explain the need to look at ancestral origins of OCA mutations.

Author response to reviewer #1 comment #3. We have completely re-written the Introduction and Discussion to provide better context on why we are investigating mutations of TYR and OCA2 genes. As suggested, we have clearly specific our hypothesis and research objective in the revised manuscript. 

Reviewer #1 comment #4. This study builds on previous research done by the team, however inclusion of this data (in form of tables) needs to be incorporated and background work needs to be elaborated to build connections to present study (Page 5; 85-94; 106).

Author response to reviewer #1 comment #4. As suggested, we have worked to clarify how the findings of our previous work on the genetics of OCA, and the work of others in the field, informs the current study. In particular, we clarify that OCA-causing mutations were characterized in previous studies, whereas we are inferring the ancestral origins of the haplotypes on which these mutations reside for our current study (see revised Introduction and Table 1 for mutations characterize from previous work together with ancestry inferences from this study). 

Reviewer #1 comment #5. What are inbreeding estimates of the study populations? Authors should provide relevant data to support the statement that “By examining population migration in the Andes regions and ancestral origins of the described mutations we report how both inbreeding and outbreeding increase the risk to develop autosomal recessive traits.” (Page 6; 111-113)

Author response to reviewer #1 comment #5. We have simultaneously tried to tone down the language on inbreeding and outbreeding, while also providing explicit support for any assertions related to these concepts from our own findings (eg Figure 3 and Table 1) and previous studies. We do not discuss inbreeding or outbreeding in the revised Introduction. Rather, we only bring in these concepts in the Discussion as a theoretical framing of our results (see Discussion page 11 paragraph 3). We do provide a citation to a previous study on inbreeding coefficient in the Marinilla-Santuario region, but we also mention here that “However, given the relatively low sample size of two Marinilla-Santuario samples, and the lack of pedigree linking these samples, this conclusion is not as well supported as the G47D founder mutation.” (see Discussion, page 12, paragraph 2).

Results

Reviewer #1 comment #6. Page 6; 116-118 – PCA analysis and global ancestry analysis should be explained for study populations.

Author response to reviewer #1 comment #6. As suggested, we provide additional details on PCA in the Methods section on Genetic ancestry inference (see Methods, page 6, first paragraph) and the Results section on Genetic ancestry and admixture of Colombian OCA patients (see Results, page 6, second paragraph). 

Reviewer #1 comment #7. The methods focus on haplotype comparisons using pedigree. The choice and justification of this methodological approach should be highlighted.

Author response to reviewer #1 comment #7. We have rewritten the Introduction, Methods, and Results sections to clarify that we use haplotype analysis in order to infer the ancestral origins of TYR and OCA2 mutations. 

Reviewer #1 comment #8. Results to be sectioned based on analysis. Detailed results to be explained for the pedigree and transmission of OCA mutations.

Author response to reviewer #1 comment #8. As suggested, in the revised manuscript we clarify exactly what the pedigree allows us to observe, ie the inheritance patterns on TYR mutations together with their ancestral origins in the Results (see page 10, first paragraph) and in the Discussion (see page 11, third paragraph and page 12, third paragraph). 

Reviewer #1 comment #9. Role of inbreeding is not measured nor indicated in results as stated in Introduction

Author response to reviewer #1 comment #9. We have simultaneously tried to tone down the language on inbreeding and outbreeding, while also providing explicit support for any assertions related to these concepts from our own findings (eg Figure 3 and Table 1) and previous studies. We do not discuss inbreeding or outbreeding in the revised Introduction. Rather, we only bring in these concepts in the Discussion as a theoretical framing of our results (see Discussion page 11 paragraph 3). We do provide a citation to a previous study on inbreeding coefficient in the Marinilla-Santuario region, but we also mention here that “However, given the relatively low sample size of two Marinilla-Santuario samples, and the lack of pedigree linking these samples, this conclusion is not as well supported as the G47D founder mutation.” (see Discussion, page 12, paragraph 2).

Discussion

Reviewer #1 comment #10. Page 8; 147-150 - This study is not investigating the complex demographic history of region but is trying to infer patterns of 2 gene mutations in the context of the complex demographic history of Columbia.

Author response to reviewer #1 comment #10. We agree. We have removed this statement and completely rewritten the Discussion to focus on what we actually did in the paper, ie analyze the ancestral origins of TYR and OCA2 mutations.

Reviewer #1 comment #11. A large part of the discussion focuses on history and ancestry of these populations and region. While this is essential, authors should focus on genes under study and discuss their results in context of prior work and population history of the region.

Author response to reviewer #1 comment #11. We agree. We have removed much of this unnecessary material from the revised Discussion, which now focuses on the genes under study and the ancestral origins of the mutations in the context of prior work and the population history of the region. 

Reviewer #1 comment #12. Page 10; 181-183 – is there evidence of homozygosity due to inbreeding for other loci from this region to support this statement?

Author response to reviewer #1 comment #12. We do have evidence from the phylogeny in Figure 2 and the pedigree shown in Figure 3 that the same haplotype was inherited from both parents for two affected individuals, pointing to some degree of consanguinity. Nevertheless, as stated in response to other comments on this same point we have simultaneously tried to tone down the language on inbreeding and outbreeding, while also providing explicit support for any assertions related to these concepts from our own findings (eg Figure 3 and Table 1) and previous studies. We do not discuss inbreeding or outbreeding in the revised Introduction. Rather, we only bring in these concepts in the Discussion as a theoretical framing of our results (see Discussion page 11 paragraph 3). We do provide a citation to a previous study on inbreeding coefficient in the Marinilla-Santuario region, but we also mention here that “However, given the relatively low sample size of two Marinilla-Santuario samples, and the lack of pedigree linking these samples, this conclusion is not as well supported as the G47D founder mutation.” (see Discussion, page 12, paragraph 2).

Reviewer #1 comment #13. Page 10; 184-186 – Authors should provide reference of estimates of genetic diversity of Bogotá, the capital of Colombia.

Author response to reviewer #1 comment #13. We have eliminated any unsupported statements on genetic diversity as suggested. Please see our response to the above comment #12 as well.

Reviewer #1 comment #14. Author’s proposal that compound heterozygotes found in individuals who carry mutations derived from varied ancestral sources can be due to outbreeding needs support from other population genetics studies. It may be useful to use population genetic models to explain how inbreeding and outbreeding affect compound heterozygotes.

Author response to reviewer #1 comment #14. On the one hand, our revised manuscript now shows clear evidence of admixture-derived compound heterozygotes, and we hope that this now comes across much more clearly. On the other hand, as stated in response to previous comments, we have completely rewritten the Introduction and Discussion to tone down the discussion of inbreeding and outbreeding. We only mention this now in the Discussion where we have explicit support from our own results and we tie this directly to previous theoretical literature on the subject (see Discussion page 11 paragraph 3, and page 12 paragraphs 2 & 3). 

Methodology

Reviewer #1 comment #15. Samples size is too limited to perform robust statistical analysis.

Author response to reviewer #1 comment #15. In our revised manuscript, we now clarify that our study aims to provide a descriptive analysis of the ancestral origins of the TRY and OCA2 mutations. Despite the low sample size of our study cohort, the large amount of genome-wide variant data that we characterized (~720k variants) and the large number of global reference samples that we used (515) provide sufficient resolution for us to achieve this objective. New details on these numbers can be found in the revised Methods section on Genetic ancestry inference (see page 6, paragraph 1). 

Reviewer #1 comment #16. Methodological choice of taking pedigree data needs to be justified.

Author response to reviewer #1 comment #16. In the revised manuscript, we more clearly illustrate the utility of the pedigree to simultaneously illustrate the inheritance patterns of TYR mutations in a Colombian family and their ancestral origins (see Results, page 10, paragraph 1; Discussion, page 11, paragraph 3 and page 12, paragraph 3).

Reviewer #1 comment #17. Authors should include a brief description of study participants and their recruitment here for clarity and not just cite the previous study.

Author response to reviewer #1 comment #17. As suggested, we provide additional details on study participants and their recruitment in the revised Methods section on Study cohort and ethics (see page 4, paragraph 3 to page 5, paragraph 1) 

Reviewer #1 comment #18. How was the recruitment done for non-related individuals and what were ethical considerations while recruiting them.

Author response to reviewer #1 comment #18. As suggested, we provide additional details on participant recruitment and all ethical considerations in the revised Methods section on Study cohort and ethics (see page 4, paragraph 3 to page 5, paragraph 1)

---

## [Decision Letter · Decision Letter 1]

5 Jul 2024

PONE-D-23-39938R1Ancestral origins of TYR and OCA2 gene mutations in oculocutaneous albinism from two admixed populations in ColombiaPLOS ONE

Dear Dr. Nunez-Rios,

Thank you for submitting your manuscript to PLOS ONE. After careful consideration, we feel that it has merit but does not fully meet PLOS ONE’s publication criteria as it currently stands. Therefore, we invite you to submit a revised version of the manuscript that addresses the points raised during the review process.

Adhere to the points made by reviewer 1.

In particular the restructuring of the discussion.

See also our minor comments.

We look forward to receiving your revised manuscript.

Kind regards,

Arnar Palsson, Ph.D.

Academic Editor

PLOS ONE

Journal Requirements:

**Additional Editor Comments:**

Adhere to the points made by reviewer 1.

In particular the restructuring of the discussion.

Minor comments.

Change “However, an increasing number of compound heterozygotes are found in diverse,

admixed populations.” To “Due to methodological advances, an increasing number of compound heterozygotes have been found including in diverse, admixed populations.”

Change “ancestry compared to low levels of African ancestry” to “ancestry, and relatively low levels of African ancestry”

After “sampled from this region [12].” Insert a linking sentence about admixture and genetic diseases, citing relevant literture/reviews.

Reword “introduced OCA causing mutations with distinct

ancestral origins onto the same genetic background in individuals from the Andean region”

maybe say “mutation of distinct origins” and skip the “genetic background”?

Skip “mutated” in “found in 21 out of the 24 mutated TYR haplotypes among”

This sentence is unclear to me “The individual with the second unknown mutated TYR haplotype is likely to be G47D homozygous based on the Spanish origin of that second haplotype.”

This argument does not hold. Second haplytype G47D is ok, but why should the second haplotype be G47D?? Point to individual in Figure or use identifier. Reword this and clarify.

Im sure more ethnicities are underrepresented – thus “being particularly underrepresented [33-35].” Is not an accurate statement. Start more generally, many populations are underrepresented in genetic research, including the Latin americans.

Shorten “for studies like this one aimed at” to “for studies aimed at”

Reviewers' comments:

Reviewer's Responses to Questions

**Comments to the Author**

1. If the authors have adequately addressed your comments raised in a previous round of review and you feel that this manuscript is now acceptable for publication, you may indicate that here to bypass the “Comments to the Author” section, enter your conflict of interest statement in the “Confidential to Editor” section, and submit your "Accept" recommendation.

Reviewer #1: All comments have been addressed

2. Is the manuscript technically sound, and do the data support the conclusions?

Reviewer #1: Yes

3. Has the statistical analysis been performed appropriately and rigorously? 

Reviewer #1: Yes

4. Have the authors made all data underlying the findings in their manuscript fully available?

Reviewer #1: Yes

5. Is the manuscript presented in an intelligible fashion and written in standard English?

Reviewer #1: Yes

6. Review Comments to the Author

Reviewer #1: 1. Authors have addressed most of the concerned raised and have rewritten the manuscript.

2. Methods section needs more detailing with regard to analysis conducted eg. pedigree analysis. Results mentions data from pedigrees, however methods section does not mention pedigree at all.

3. Discussion needs to be reworked based on the suggestions given.

7. PLOS authors have the option to publish the peer review history of their article (what does this mean?). If published, this will include your full peer review and any attached files.

Reviewer #1: **Yes: **Manjari Jonnalagadda

---

## [Author Response · Author response to Decision Letter 1]

21 Aug 2024

Manuscript Title: Ancestral origins of TYR and OCA2 gene mutations in Oculocutaneous albinism 2 from two admixed populations in Colombia

Dear Editor and reviewers, we are particularly grateful for your valuable comments. We detail the changes made in response to the editor and reviewer comments below, indicating how and where we changed the manuscript in response to each point.

Introduction – 

1. “We are interested in the relationship between genetic ancestry, admixture, and the presence of mutations that cause oculocutaneous albinism (OCA) in Colombian populations from the Andean region.” – Why? can authors state why this trait (and thereby genes OCA2 and TYR) is of interest to them? Is there a phenotypic association of significance in this population?

Response: We have added information to the abstract, introduction, and discussion to clarify our interest in this trait. Specifically, we explained that this study builds on our previous findings, aiming to explore the ancestral origins of reported mutations in the context of Colombia's complex colonization history. Understanding these origins is critical for deciphering the genetic basis of OCA in this region, where unique admixture patterns may influence disease prevalence.

2. The specific objective of this study was to test our admixture-derived compound

heterozygote hypothesis by characterizing the ancestral origins of TYR mutations from Altiplano Cundiboyacense and OCA2 mutations from the Marinilla-Santuario region in Antioquia” - The proposed hypothesis - admixture-derived compound heterozygote hypothesis has support from a previous study looking at the G47D mutation in OCA patients living in the Canary Islands and Puerto Rico and suggested to have Sephardic Jewish origin based on its high prevalence among Moroccan Jews living in Israel [23]. If the authors are retesting this hypothesis in the Andean population, they need to explain the changed/varied context that demands retesting of the hypothesis. It could be as simple as effect of these heterozygote mutations on this population. 

Response: We have expanded our explanation to clarify that the origins of mutations found in OCA patients from Colombia remain uncertain. Considering the multiple waves of immigration and complex admixture in Colombia, along with diversity in genetic backgrounds that can lead to unique combinations of mutations that might not be observed in more genetically homogeneous populations, we aimed to examine the ancestral origins of these mutations and then correlate those findings with inbreeding and outbreeding depression in the examined regions. 

3. “We characterized the genome-wide patterns of genetic ancestry for OCA patients from Altiplano Cundiboyacense and Antioquia, confirming European and Native American admixture with little African ancestry.. Our results underscore the genetic complexity of Mendelian disease in admixed Latin American populations” – this is a part of this study? The placement of the paragraph is odd as part of the introduction. It is suggested to include this section as part of the introduction to the discussion.

Response: We have removed the mentioned paragraph from the introduction and integrated the relevant information into the results and discussion sections

Results 

– “The OCA1 patients studied here were sampled from in …. or near Altiplano Cundiboyacense, primarily in the departments of Cundinamarca and Boyacá (Figure 1A) …. genomic variant data from African, European, and Native American reference populations.” – This content is repeated. It’s already included in Methods Study cohort section.

Response: This information has been removed from the results section and was appropriately integrated to avoid redundancy.

4. “Principal component analysis (PCA) shows that the Colombian OCA patients studied here group closely together and fall between the… principal component 2

(PC2; Figure 1B) - statement can be changed to PCA was performed to evaluate… and PCA shows ….

Response: We have revised this section to reflect the suggestion. The methods paragraph describing the PCA was removed, and the results were rephrased to clarify that PCA was performed to evaluate genetic clustering, and the results were described accordingly.

5. “The pedigree in Figure 3 shows the inheritance patterns and ancestral origins for OCA1 causing

TYR mutations…. – Firstly, acquisition of Pedigree data (why, how) doesn’t find any mention in the methods section. Pedigree data is crucial to testing the admixture-derived compound heterozygote hypothesis, it must be included in the Methods section.

Response: We have added a detailed explanation of the acquisition and analysis of pedigree data in the Methods section, highlighting that this study builds on our previous findings from our group and the same nomenclature is preserved here. 

Authors can consider the following points to revise the discussion.

1. The discussion can begin with “Our study on the genetic etiology of OCA in Colombia highlights the valuable insights that can be obtained from clinical genomics studies of Latin American populations characterized by significant genetic admixture…..Andean region, where our study populations live, is characterized by two-way European and Native American ancestry, including a previously hidden Converso ancestry component (Figure 1) [12].” 

Response: The discussion has been rewritten to follow a more logical flow, beginning with a broader overview of our study's significance in the context of genetic studies of Latin American populations, then moving to specific findings from our study.

2. Why the focus on OCA?

Response: This suggestion has been incorporated along the manuscript 

3. Summarize the key findings here and explain their patterns in context of admixture and ancestry of study populations and highlight ancestry profile of Columbians in general. Highlight previous studies on genetic etiology of OCA in Colombia characterized by significant genetic admixture and OCA (OCA1 and OCA2) frequencies in Columbian populations if any. If not, states there aren’t any other studies investigating this.

Response: We have expanded the discussion to include a summary of key findings, contextualizing them within the broader ancestry profile of Colombians. We also highlighted the significance of our previous studies and noted the lack of comprehensive data on OCA frequencies in Colombian populations, underscoring the importance of this research.

4. Explain Pedigree data and draw parallels with other diseases studied in these populations.

Response: We added a comparison to other studies conducted in the same regions, reinforcing the idea that the genetic complexity resulting from admixture increases the likelihood of inherited disorders due to compound heterozygosity.

---

## [Decision Letter · Decision Letter 2]

10 Oct 2024

PONE-D-23-39938R2Ancestral origins of TYR and OCA2 gene mutations in oculocutaneous albinism from two admixed populations in ColombiaPLOS ONE

Dear Dr. Nunez-Rios,

Thank you for submitting your manuscript to PLOS ONE. After careful consideration, we feel that it has merit but does not fully meet PLOS ONE’s publication criteria as it currently stands. Therefore, we invite you to submit a revised version of the manuscript that addresses the points raised during the review process.

The reviewers gave the green light on the manuscript but I found a few minor things that I want you to fix before acceptance.

Comment

First off, it is not helpful that the manuscript did not come with line numbers, that the “track changes“ version was nearly all read that there was a third version of the manucript in the submission package. Please proved a better “track changes“ version  for the final submission.

Figure 1B is unclear as some individuals sit on top of one another in the PC plot. Redo the figure either with smaller symbols or some level of transparency.

Figure 2 legend (and maybe result text). How long are the regions used to determine the haplotype ancestry around TYR? The genomic region in kb and perhaps also the nr of SNP used to define their ancestry.

The verb “characterize“ is used liberally. Below I suggest some place where it can be dropped. Please skim through and check other places where simpler verbs may suffice.

Minor comments.

We look forward to receiving your revised manuscript.

Kind regards,

Arnar Palsson, Ph.D.

Academic Editor

PLOS ONE

Journal Requirements:

Additional Editor Comments (if provided):

The reviewers gave the green light on the manuscript but I found a few minor things that I want you to fix before acceptance.

Comment

First off, it is not helpful that the manuscript did not come with line numbers, that the “track changes“ version was nearly all read that there was a third version of the manucript in the submission package. Please proved a better “track changes“ version for the final submission.

Figure 1B is unclear as some individuals sit on top of one another in the PC plot. Redo the figure either with smaller symbols or some level of transparency.

Figure 2 legend (and maybe result text). How long are the regions used to determine the haplotype ancestry around TYR? The genomic region in kb and perhaps also the nr of SNP used to define their ancestry.

The verb “characterize“ is used liberally. Below I suggest some place where it can be dropped. Please skim through and check other places where simpler verbs may suffice.

Minor comments.

Page 4 line 5-

Replace“ may have introduced OCA causing mutations with distinct

ancestral origins onto the same genetic background in individuals from the Andean region” with “may have lead to multiple OCA mutations of distinct ancesries segregating in individuals in the Andean region”

Page 4 line 7.

Provide a citation for the “admixture-derived compound heterozygote hypothesis” from the literature.

Page 4, paragraph 3

Replace “Our findings demonstrate that the G47D mutation is present on a Spanish-origin haplotype and not a Sephardic Jewish” with

“The data demonstrate that the G47D mutation sits on on a Spanish-origin haplotype and not a Sephardic Jewish…”

Page 5, paragraph 3

Replace “as described for the previous studies where these samples were first reported [21, 22]” with “as described in previous studies on these individuals [21, 22]”

Page 5, paragraph 4

Replace “The same samples were characterized for this study with” with “These individuals were genotyped with”

Page 5, paragraph 4, line 8

Remove “taken”

Page 6, line 1.

Reword “local (i.e. haplotype-specific) ancestry inference”

To “local (i.e. haplotype around the TYR and OCA genes) ancestry inference”

Page 11, paragraph 2, line 2

Replace “Latin American populations characterized by significant genetic” to “Latin American populations with significant genetic”

Page 11, paragraph 2, line 5

Replace “The Colombian population is characterized by African, European” to “The Colombian population has African, European”

Page 11, paragraph 2, line 8.

Reword “The Andean region, where our study populations live, is characterized by two-way…” to “The study population in the Andean region mainly has European and Native American ancestry, and as the data revealed a notable Converso ancestry component”

Page 11, paragraph 3, line 4

Reword “are homozygous with copies of the same G47D haplotype” to

“are homozygous for the same G47D haplotype”

Page 12, paragraph 2, line 3

Reword “A previous study found that the Marinilla-Santuario region had the highest inbreeding coefficient in Colombia [42], consistent with the idea that the A787T mutation also

represents a founder effect.” To “This observation and the previous finding that the Marinilla-Santuario region had the highest inbreeding coefficient in Colombia [42], is consistent with the idea that a founder event influenced variation in this region in particular”

Reviewers' comments:

Reviewer's Responses to Questions

**Comments to the Author**

1. If the authors have adequately addressed your comments raised in a previous round of review and you feel that this manuscript is now acceptable for publication, you may indicate that here to bypass the “Comments to the Author” section, enter your conflict of interest statement in the “Confidential to Editor” section, and submit your "Accept" recommendation.

Reviewer #1: All comments have been addressed

2. Is the manuscript technically sound, and do the data support the conclusions?

Reviewer #1: Partly

3. Has the statistical analysis been performed appropriately and rigorously? 

Reviewer #1: Yes

4. Have the authors made all data underlying the findings in their manuscript fully available?

Reviewer #1: Yes

5. Is the manuscript presented in an intelligible fashion and written in standard English?

Reviewer #1: Yes

6. Review Comments to the Author

Reviewer #1: Authors have addressed the concerns raised and provided necessary changes to make the manuscript a better read.

7. PLOS authors have the option to publish the peer review history of their article (what does this mean?). If published, this will include your full peer review and any attached files.

Reviewer #1: **Yes: **Manjari Jonnalagadda

---

## [Author Response · Author response to Decision Letter 2]

29 Oct 2024

We appreciate the reviewers' insightful comments and suggestions, which have been invaluable in refining our manuscript. In response, we have carefully addressed each point and made corresponding changes, as outlined in detail below

1. First off, it is not helpful that the manuscript did not come with line numbers, that the “track changes“ version was nearly all read that there was a third version of the manuscript in the submission package. Please proved a better “track changes“ version for the final submission.

Changes made to the manuscript were implemented as follows:

a) In the first round of corrections, the manuscript was revised by a native English speaker. All edited paragraphs are marked in red.

b) In the second round of corrections, track changes were used to document revisions.

c) In the third round of corrections, track changes were again utilized, with all modifications highlighted in yellow.

2. Figure 1B is unclear as some individuals sit on top of one another in the PC plot. Redo the figure either with smaller symbols or some level of transparency.

Figure 1B was corrected with some level of transparency.

3. Figure 2 legend (and maybe result text). How long are the regions used to determine the haplotype ancestry around TYR? The genomic region in kb and perhaps also the nr of SNP used to define their ancestry.

Information was added in methods text. 

4. The verb “characterize“ is used liberally. Below I suggest some place where it can be dropped. Please skim through and check other places where simpler verbs may suffice. 

We thank you the reviewers for those suggestions. All changes were highlighted in yellow 

Minor comments.

a. Page 4 line 5-

Replace“ may have introduced OCA causing mutations with distinct

ancestral origins onto the same genetic background in individuals from the Andean region” with “may have lead to multiple OCA mutations of distinct ancesries segregating in individuals in the Andean region”

Completed

b. Page 4 line 7.

Provide a citation for the “admixture-derived compound heterozygote hypothesis” from the literature.

Citations were added

c. Page 4, paragraph 3

Replace “Our findings demonstrate that the G47D mutation is present on a Spanish-origin haplotype and not a Sephardic Jewish” with

“The data demonstrate that the G47D mutation sits on on a Spanish-origin haplotype and not a Sephardic Jewish…”

This sentence was removed in the second round of corrections. No additional changes were conducted in this round.

d. Page 5, paragraph 3

Replace “as described for the previous studies where these samples were first reported [21, 22]” with “as described in previous studies on these individuals [21, 22]”

Completed

e. Page 5, paragraph 4

Replace “The same samples were characterized for this study with” with “These individuals were genotyped with”

Completed

f. Page 5, paragraph 4, line 8 Remove “taken”

Completed

g. Page 6, line 1.

Reword “local (i.e. haplotype-specific) ancestry inference”

To “local (i.e. haplotype around the TYR and OCA genes) ancestry inference”

Completed

h. Page 11, paragraph 2, line 2

Replace “Latin American populations characterized by significant genetic” to “Latin American populations with significant genetic”

Completed

i. Page 11, paragraph 2, line 5

Replace “The Colombian population is characterized by African, European” to “The Colombian population has African, European”

Completed

j. Page 11, paragraph 2, line 8.

Reword “The Andean region, where our study populations live, is characterized by two-way…” to “The study population in the Andean region mainly has European and Native American ancestry, and as the data revealed a notable Converso ancestry component”

Completed

k. Page 11, paragraph 3, line 4

Reword “are homozygous with copies of the same G47D haplotype” to

“are homozygous for the same G47D haplotype”

Completed

l. Page 12, paragraph 2, line 3

Reword “A previous study found that the Marinilla-Santuario region had the highest inbreeding coefficient in Colombia [42], consistent with the idea that the A787T mutation also

represents a founder effect.” To “This observation and the previous finding that the Marinilla-Santuario region had the highest inbreeding coefficient in Colombia [42], is consistent with the idea that a founder event influenced variation in this region in particular”

Completed

---

## [Editor Report · Decision Letter 3]

31 Oct 2024

Ancestral origins of TYR and OCA2 gene mutations in oculocutaneous albinism from two admixed populations in Colombia

PONE-D-23-39938R3

Dear Dr. Nunez-Rios,

We’re pleased to inform you that your manuscript has been judged scientifically suitable for publication and will be formally accepted for publication once it meets all outstanding technical requirements.

Kind regards,

Arnar Palsson, Ph.D.

Academic Editor

PLOS ONE
---

## [Editor Report · Acceptance letter]

7 Nov 2024

PONE-D-23-39938R3 

PLOS ONE

Dear Dr. Nunez-Rios, 

I'm pleased to inform you that your manuscript has been deemed suitable for publication in PLOS ONE. Congratulations! Your manuscript is now being handed over to our production team.

Kind regards, 

on behalf of

Dr. Arnar Palsson 

Academic Editor

PLOS ONE